# A Critical Overview of the Construct of Supportive Care Need in the Cancer Literature: Definitions, Measures, Interventions and Future Directions for Research

**DOI:** 10.3390/ijerph21020215

**Published:** 2024-02-12

**Authors:** Marco Miniotti, Rossana Botto, Giovanna Soro, Alberto Olivero, Paolo Leombruni

**Affiliations:** ‘Rita Levi Montalcini’ Department of Neuroscience, University of Turin, 10126 Turin, Italy; rossana.botto@unito.it (R.B.); giovanna.soro@unito.it (G.S.); alberto.olivero@unito.it (A.O.); paolo.leombruni@unito.it (P.L.)

**Keywords:** supportive care needs, cancer, psychosocial oncology, patient-reported outcomes, assessment

## Abstract

The growing amount of evidence about the role of supportive care in enhancing cancer patients’ outcomes has made healthcare providers more sensitive to the need for support that they experience during cancer’s trajectory. However, the lack of a consensus in the definition of supportive care and lack of uniformity in the theoretical paradigm and measurement tools for unmet needs does not allow for defined guidelines for evidence-based best practices that are universally accepted. Contemporary cancer literature confirms that patients continue to report high levels of unmet supportive care needs and documents the low effectiveness of most of the interventions proposed to date. The aim of this critical review is to consolidate the conceptual understanding of the need for supportive care, providing definitions, areas of expertise and a careful overview of the measurement tools and intervention proposals developed to date. The possible reasons why the currently developed interventions do not seem to be able to meet the needs, and the issues for future research were discussed.

## 1. Introduction

Cancer can be conceptualized as an unexpected life event or a series of events across the illness’ trajectory. For some, cancer can be experienced as a single event, with a defined beginning and ending. For others, cancer can become a constant in their lives. For all, cancer involves entry into a cancer care system, each one from different points, with different paces, along different paths. This consists of a complex experience of events, reactions and interactions, from diagnostic investigation through to diagnosis, treatment, rehabilitation, follow-up and then, survival for some, recurrence and death for the others, that become embedded in daily life. Illness-related demands take time away from life, hinder purposes and businesses, interfere with plan- and decision-making and alter relational and intimate balances, with negative sequelae on family members’ physical and mental health. Moreover, when people are facing a life-threatening illness, thoughts about the meaning in life, mortality and the afterlife typically arise [1,2].

People living with cancer must address a broad spectrum of changes and challenges, as the disease puts a strain on the resilience of the person in every way. As cancer occurs, patients must face physical symptoms, such as weight loss, fatigue and pain. Moreover, a cancer diagnosis is usually associated with emotional disturbances such as anxiety and a depressed mood and are also associated with a sense of a personal crisis and the perception of a biographical disruption [1,2,3]. When active treatments begin, the patients start to experience a wide range of side effects and adverse events. Surgery exposes them to postoperative debilitation, pain and dissatisfaction with their body image. Chemotherapy triggers toxicity responses and causes fatigue, nausea and vomiting, taste alteration, widespread pain, sleep disturbances, febrile neutropenia and aplastic anemia. Radiotherapy results in localized inflammatory reactions, skin changes, fatigue and chronic fibroses. Immunotherapy prompts autoimmune reactions, causing skin rashes, colitis, nephritis, hepatitis, pneumonia and endocrinal comorbidities [4,5,6]. As curative treatments end, patients are faced with follow-up uncertainties and challenges, and may develop fatigue and anxiety due to continuous screenings, have a constant fear of recurrence and feel burdened with multiple complications and comorbidities related to cancer chronicization, all while fighting for survival. Lastly, patients with advanced cancer generally experience states of profound physical, emotional and existential suffering as they approach death [7,8].

Coping with the arising demand imposed by the cancer journey can be overwhelming for patients and their families, and emotional distress, psychological disturbances, psychiatric disorders and existential suffering can be considered expected responses to illness [3]. As individuals or nodes in a network of relationships and interactions, cancer patients and their family members are human beings, and as such, they act each day to ensure their needs are fulfilled, whether physical, emotional, informational, practical, social, existential or spiritual. The needs can differ from person to person, and the same person can experience different needs at different stages of the disease, as the cancer unfolds and treatment changes [1]. Whether or not a person will be able to meet his/her own needs by him/herself depends on a variety of factors and variables. Symptom and treatment burdens, the loss of functioning and disability, emotional suffering and the emergence of death anxieties may interfere with the person’s ability to act and cope, and then, feelings of dejection, disheartenment, hopelessness, helplessness, loneliness, and losses of meaning and purpose in life may emerge into the person’s internal experience, with detrimental effects on the person’s psychological adjustment to the disease [1,2,3]. In such a psychological condition, even asking for and receiving help can be difficult and even beyond the person’s concrete possibilities. The frustration resulting from the inability to fulfil their own needs and the perception of the lack of support from others may further threaten the loss of mastery and control, which are already challenged by the progression of the disease, representing an additional burden that weighs on the experiences of cancer patients and their family members.

To prevent cancer patients and their families from being overwhelmed and to help them finding the best physical and psychological adjustments within their reach, nowadays, cancer care pathways and facilities involve multiple disciplines and professionals who collaborate with each other as a part of an interdisciplinary team appointed to support patients in meeting their needs, so as to positively impact their psychophysical wellbeing, health-related quality of life [9,10] and survival outcomes [11]. It is customary to refer to this kind of complex approach to care with the term ‘supportive care’.

## 2. Definitions and Conceptualizations of Need for Supportive Care

The term ‘supportive care need’ first emerged in the 1980s, but became more and more used in the cancer literature during the 1990s [12]. In 1994, Fitch, among the first, defined supportive care as “…the provision of the necessary services for those living with or affected by cancer to meet their physical, emotional, social, psychological, informational, spiritual and practical needs during the diagnostic, treatment, and follow-up phases, encompassing issues of survivorship, palliative care and bereavement” [13]. This definition clarifies that supportive care is devoted to meet the needs of both the patients and caregivers throughout the whole cancer trajectory, including the prevention, diagnosis, active treatment, follow-up, survivorship and bereavement, encompassing a broad range of contexts and fields (i.e., physical, emotional, spiritual, social and informational).

Currently, the term ‘supportive care need’ is commonly used in oncology, but a 2013 systematic review of the cancer literature documented such a wide heterogeneity in how it has been used, that it risks creating confusion for clinicians and patients alike [14]. The expression ‘supportive care need’ can be considered an ‘umbrella term’, i.e., a term used to cover a wide-ranging subject, assuming different meanings and practical connotations depending on the setting or also, as in this case, the population, the stage or treatment one refers to [15]. In general, ‘supportive care need’ refers to the need to receive help, care and assistance with respect to a specific issue that is causing discomfort and suffering to a cancer patient. The need may result from suffering or limitations caused by physical symptoms associated with the disease and treatment, which in turn may arouse issues and concerns in facing daily activities, preserving roles and/or maintaining acceptable levels of stability and wellbeing. Needs may be inherent in the psychological, relational or spiritual sphere, whose healthy balance can be altered by fears and anguish toward cancer progression/recurrence or by the loss of meaning and purpose in life. The need may even be caused by having to deal with pain, losses or the threat of death, or sustained by the difficulty in understanding medical terms related to one’s illness, technical aspects of the interventions and treatments one must undergo, their medical reasons and related benefits or side effects.

Therefore, when requesting supportive care for a patient, it would be appropriate to specify (even to the patient in return) the context in which the need arose and its domain of relevance, in order to welcome discomfort empathically and manage the need adequately.

To facilitate a good understanding of patients’ supportive care needs, Hui classified supportive care into three levels based on the level of specialization requested [12]: Primary supportive care is generally provided by health professionals in an outpatient setting (e.g., oncologists and nurses), and entails, for instance, responses to the need for understanding the diagnosis/prognosis, prophylaxis and management of the cancer/treatment-related physical burden. Secondary supportive care is generally provided by specialized teams via inpatient consultations, and concerns, for instance, responses to the needs regarding medical or psychological comorbidities. Tertiary supportive care is generally provided by teams specialized in managing the multifaceted needs of patients with advanced disease or those close to the end-of-life through highly complex interventions (e.g., palliative sedation), delivered in palliative care units or hospices. This categorization is appreciable because it restores order between expressions frequently used interchangeably, such as ‘standard care’, ‘supportive care’, palliative care’ and ‘hospice care’. It follows such that when primary supportive care is delivered by oncologists and nurses, this service can be considered standard care, whereas services devoted to supportive care address the needs concerning both survivors and dying patients; thus, it may be secondary or tertiary supportive care and may or may not include palliative care or hospice programs. Conversely, palliative care and hospice programs always fall within tertiary supportive care interventions.

## 3. Measures of Need for Supportive Care

To assess patients’ morbidity outcomes, three parameters have been considered in the cancer literature: first, the health-related quality of life and satisfaction for care, and more recently, the need for supportive care [16]. Unlike the primary measures of quality of life and satisfaction for care, a needs assessment offers a number of documented advantages, as this approach provides a direct measure of the patient’s perceived need for help with respect to a number of common concrete situations for generic or peculiar cancer patients, providing a personal magnitude of the need, regardless of other aspects of life experience and quality of care [16].

Starting in the mid-1990s and to a greater extent, from the 2000s onwards, a number of tools have been developed to assess the needs of the cancer patient population. To date, 17 instruments have been released in the literature (see Table 1). The Cancer Rehabilitation Evaluation System (CARES) assesses the needs related to post-treatment rehabilitation and daily activities [17]. A few years later, the 76-item Cancer Patients Needs Questionnaire (CPNQ) was developed [18]; this tool has a shorter form [19] and a version for breast cancer [20] and for parents of sons with cancer [21]. The most widely used, adapted and translated worldwide to date tool is the Supportive Care Needs Survey (SCNS), based on the experience of the CPNQ, originally developed as a 59-item questionnaire (SCNS-LF59) and later reduced to a 34-item scale (SCNS-SF34) [16,22]. The SCNS-SF34 proved to be a psychometrically valid, reliable and invariant tool in a large number of studies, resulting in numerous language translations and cultural adaptations [23,24,25,26,27,28], as well as specific versions for partners/caregivers [29] and for screening purposes [30]. The SCNS-SF34 has additional individually validated modules for specific cancer sites (i.e., breast, head and neck, prostate, colon and melanoma) [22]. The Cancer Needs Distress Inventory (CaNDI) assesses the distress that arose from having unmet needs related to the cancer experience from a biopsychosocial perspective [31]. The Cancer Survivors’ Unmet Needs (CaSUN) [32], its cultural adaptations [33,34] and its version for patients’ partners (CaSPUN) [35] assess the needs of survivors. The Survivor Unmet Needs Survey (SUNS) [36], its short form (SUNS-SF) [37] and its version for hematologic cancers (SUNS-hematological) [38] target survivors 1 to 5 years after diagnosis. The Need Assessment for Advanced Cancer Patients (NA-ACP) [39], the Problems and Needs in Palliative Care [40] and the Simple Screening Tool for Identifying Unmet Palliative Care Needs (SST-IUPCN) [41] questionnaires and their shortened versions [42,43] assess the supportive care needs of advanced cancer patients in palliative care. The Screen for Palliative and End-of-Life care needs in the Emergency Department (SPEED) [44] evaluates the palliative care needs of cancer patients presenting to the emergency department. The Three-Levels-of-Needs (3LNQ) [45] is a supplementary module of the European Organization for Research and Treatment of Cancer Quality of Life Questionnaire (EORTC QLQ-C30) used to assess supportive palliative care needs. The Information Styles Questionnaire (ISQ) [46] assesses the need to be informed about the medical care of cancer patients. The Needs Evaluation Questionnaire (NEQ) [47] and its version for outpatients [48] assess the main emotional and practical care needs of cancer patients, similarly to the Supportive Care sCore (SCC) [49]. The Comprehensive Needs Assessment Tool (CNAT) [50] and its version for caregivers [51] assess the cancer patients’ and caregivers’ overall experiences of need felt throughout the whole illness trajectory.

## 4. Intervention Options for Unmet Supportive Care Needs

Understandably, most studies on supportive care needs focused on the peculiar unmet needs of specific cancer patient populations generally defined by their tumor site, treatment or demographics. Only quite recently, an interesting systematic review of the cancer literature synthetized the evidence on the need for supportive care across cancer patient populations, trying to identify a common set of unmet needs experienced by the majority of cancer patients [52]. The authors isolated three key groups of common needs: the illness-related work (i.e., understanding the illness and prognostic scenarios and interacting with healthcare providers effectively, with the need for information as the representative need); the everyday life work (i.e., managing illness while maintaining a sense of normalcy in life, with the need to be helped in continuing usual activities and roles as the representative need); and the biographical work (i.e., maintaining one’s identity while tackling novel issues, concerns and fears, with the need for psychological and spiritual support as the representative need). Despite the increasing attention to unmet supportive care needs overtime, the contemporary cancer literature clearly shows that patients continue to report high levels of unmet needs related to the provision of information [53,54,55,56], psychospiritual support [55,56,57,58] and practical assistance in daily living [55,59]. A previous systematic review of interventions to reduce unmet supportive care needs of cancer patients found that two thirds of the studies considered (6:9) failed to demonstrate an intervention effect on the unmet needs [60]. The authors of the review concluded that there was no evidence for any particular intervention, and that those found to be effective [61,62,63] (nurse-led or telephone-based) only produced small effects on unmet needs.

In the last decade, a series of different interventions have been proposed to meet the supportive care needs of cancer patients. For the purpose of our study, the intervention approaches have been grouped into seven categories, and a representative example has been reported and detailed for each category (see Table 2). The considered interventions ranged from the implementation of a smartphone/tablet interactive app to help patients to manage the symptom burden through self-care [64] to the development of patient education and support programs to improve coping skills [65],; telephone-based psychotherapeutic interventions to address psychological distress [66]; combined interventions for gynecological cancer, with a nurse-led consultation to address concerns about radiotherapy and peer-led telephone psychosocial support provided by gynecological cancer survivors to sustain self-care management [67]; complementary medical treatments with Swedish massage therapy to address cancer-related fatigue sequelae [68]; nurse-led, home-based interventions of light-intensity physical exercise to improve self-efficacy for fatigue self-management [69]; and the implementation of electronic health records accessible to patients to facilitate the usefulness of patient-reported outcomes [70]. The purpose, targeted population, brief description of the intervention and main results are provided in Table 2. The reported interventions do not meet all the needs for which they were developed, but they are generally effective in empowering the patient by reducing the need for support and facilitating patient-centeredness in healthcare providers.

### The Role of Psychoeducation for Both Patients and Families

Despite the cancer burden affecting the quality of life of both the patient and family caregivers, healthcare providers primarily focus their efforts on relieving patient suffering, paying less attention to the needs of family caregivers [71]. However, caregivers shared with patients the cancer-related multifaceted negative impacts on life and commonly experienced psychological and medical morbidities [72,73,74]. The research indicates that, although independent, patient and family members’ adjustments to cancer’s demands affects them both [75]. Thus, the growing body of evidence about the deleterious impact of cancer caregiving has promoted attention to family caregivers’ burdens and the need for support. Several interventional programs have been defined to provide psychoeducational support to both cancer patients and their families, with the common aim of fostering and sustaining their active participation in the care journey. Most of these interventions address the patient–caregiver dyad’s mindset and cognition toward caring and being cared for, intimate and affective dynamics, coping and communication skills, self-care strategies, distress and symptom management, healthcare system and facilities navigation and the relationship referral with healthcare providers to facilitate preferences and needs, acknowledging and thus sharing decision making and informed treatment tailoring [76,77,78,79,80,81,82]. The research demonstrated that dyadic interventions have proven more effective than individual separated interventions on patients and caregivers [83].

## 5. Conclusions and Issues for the Future

Over the past two decades, there has been a continuous increase in awareness about the importance of supportive care in enhancing the outcomes of cancer patients, and a growing attention is beginning to be placed upon their unmet needs in daily clinical practice. However, most of descriptive studies performed to date addressed topics referring to the different conceptual models of need for supportive care and assessed different outcomes through different measures, causing confusion in defining guidelines for evidence-based best practices [14].

More recent research contributions are providing a gradually recognized theoretical framework for supportive care, drawing boundaries and defining themes and areas of expertise, through which we can orient the clinical practice toward a more comprehensive cancer care [12]. However, despite these steps forward, a universally accepted, agreed-upon definition for supportive care and the unmet need for supportive care has not yet been achieved, and it could indeed be hard to achieve. The unique systematic review about interventions was written to address the gaps and lacks in supportive care for cancer patients performed to date, beyond having documented the low effectiveness of most of the proposals put forward in the literature, highlighting the clear preponderance of descriptive studies with respect to the interventional studies within supportive care need research [60]. In other words, many studies prove the problem and few offer solutions, most of which do not solve the problem. During the last decade, something seems to have changed, and some effective interventional proposals have been developed (see the previous section). However, a marked imbalance between the proposed solutions and reconfirmations of the problem in favor of the latter is still present [52]. It is therefore interesting to investigate the reason why this is so, since the dearth of effective solutions might fuel the above difficulty in defining guidelines for evidence-based practices. A possible explanation may be intrinsic to the theoretical reference paradigm. A supportive care needs assessment approach was developed to allow cancer patients to have a voice in their care; therein lies its strength, as opposed to objective measures of quality of life and satisfaction for care. However, patient-reported measures of unmet needs in cancer care might not directly reflect the gaps and lacks in the healthcare system, which, if addressed, would improve cancer patient experiences of care. Despite research, the evidence, has established links between cancer-related symptoms, loss in functioning, unmet supportive care needs and worsened quality of life and psychological distress [84,85,86,87]; there are no certainties that supportive care need measures are appropriate outcomes to evaluate the effectiveness of the intervention programs to enhance cancer care. Moreover, some psychological (e.g., death anxiety) and information (e.g., life expectancy) needs could reflect a desire for certainty that is incompatible with current medical knowledge or conceal a hope of complete healing that conflicts with reality. In these cases, it is unlikely to think of being able to intervene to extinguish those needs by providing reassurances and detailed information; the fear of dying could remain, and it could be impossible to provide certain data on how long the person will be able to live. Being aware that fear and uncertainty are common and resistant elements in the experience of many cancer patients, which continuously fuel the need to be listened to, reassured and informed about, does not mean underestimating them and surrendering to the evidence of an insurmountable obstacle toward a little well-being for people with cancer. Conversely, to stay on the needs taken as examples, many authors are currently working successfully in developing psychotherapeutic approaches to help patients manage death anxiety, others are defining increasingly precise life expectancy calculation algorithms and others are trying to set up techniques to effectively communicate this information to the patient. However, reassuring and instilling courage does not undo fear, just as communicating accurate information does not eliminate uncertainty about the future. So, fears, desires, expectations and needs are not the same thing, and it is timely to also establish a clear definition to the concept of ‘unmet need’, as it is beginning to happen for the term ‘supportive care’. Furthermore, the notable growing expansion in the development and use of supportive care need questionnaires and surveys (see ‘Measures of need for supportive care’ section) raises the primary concern about their psychometric robustness, particularly with respect to their sensitivity and specificity, and thus, their ability to distinguish responders with unmet needs from those without and appreciate changes over time. A widespread trend in the validation studies of measurement tools applied in medical sciences is to focus on the exploration of the factor structure and the investigation of the factors and scale’s internal consistencies, the test–retest reliability and the content and construct validity at most. These statistics are very informative, but particularly useful to design instruments to detect a wide breadth of issues and concerns across large populations. The questionnaires and scales developed to capture unmet needs would require an item-level analysis to establish item functioning and reliability. While generally grouped into common categories, as sums of multiple ratings, each need refers only to itself, and its resolution can only provide for a highly precise and specialized intervention on it. Many of the unmet need tools available today are built in such a way that similar scores can correspond to highly different patterns of needs; conversely, the effectiveness of a need-targeted intervention may not be detected if only the sums of ratings are considered. Beyond that, the effectiveness of an intervention is generally measured through the analysis of statistical significance or effect sizes at most. These standardized criteria fail to ascertain whether the intervention has had a real impact on patients’ perceptions of their own wellbeing. As has been achieved for tools pertaining to other research fields or for quality-of-life scales, it would also be advisable for supportive care need measures to establish criteria to define the minimal clinically important difference, i.e., the smallest difference in a score that most patients can perceive as beneficial or harmful. Proposals for identifying the clinical significance of changes in unmet need scores have not been put forward, and this is quite surprising within the supportive care need research, as this approach clearly emphasizes the primacy of a patient’s perception and is therefore fully centered on the reference theoretical paradigm. Establishing the minimal, clinically important difference values for the most used tools, the SCNS-SF34 above all, could be a primary focus for future supportive care need research, and can provide empirical evidence useful to reach a definitive consensus definition for a ‘met’ and ‘unmet’ need in supportive care. This would also equip healthcare providers with evidence-based effectiveness indicators to be considered when tailoring treatment procedures and care approaches to individual patient preferences.

Supportive care programs must offer solutions to assist and support patients and families through the full spectrum of needs that may arise throughout the cancer journey. Effective supportive care delivery requires the collaborative efforts of multiple disciplines and different professionals working collaboratively within an integrated approach orientated to patient-centered cancer care. For this to be practicable, it is necessary that curricula of healthcare profession students are enriched with educational interventions devoted to the development of attitudes toward patient-centeredness care and interprofessionalism.

## Figures and Tables

**Table 1 ijerph-21-00215-t001:** Measures options for supportive care needs.

Instruments	Domains	No. of Items	Response Options	Specificities	References
SCNS-LF59	5: psychological, health system and information, physical and daily living, patient care and support and sexuality	59	Five-point scale (1 = no need/not applicable, 5 = high need)	SCNS long form, 59 items	[16]
SCNS-SF34	5: psychological, health system and information, physical and daily living, patient care and sexuality	34	Five-point scale (1 = no need/not applicable, 5 = high need)	SCNS short form, 34 items	[22]
SCNS-SF Mandarin and Cantonese	4: health system, information and patient care, psychological, physical and daily living and sexuality	33	Five-point scale (1 = no need/not applicable, 5 = high need)	Eliminated item 19 dealing with the choice of hospital; in Hong Kong, the healthcare system is public, and patients go to the hospital closest to home because the service is equal throughout the region	[23]
SCNS-SF Mexican	5: psychological, health system and information, physical and daily living, patient care and sexuality	33	Five-point scale (1 = no need/not applicable, 5 = high need)	Eliminated item 31 "information about sexual relationships" due to high cross loadings	[24]
SCNS-SF Dutch	4: physical and psychological, hospital care, information and communication and practical and cultural needs	34	Five-point scale (1 = no need/not applicable, 5 = high need)	Redistribution of the items according to the new 4-factor structure set out by the authors	[26]
SCNS-ST9	5: psychological, health system and information, physical and daily living, patient care and support and sexuality	9	Five-point scale (1 = no need/not applicable, 5 = high need)	SCNS short form, 9 items (screening tool)	[29]
SCNS-P&C	4: information, healthcare services, daily living and psychological	44	Five-point scale (1 = no need/not applicable, 5 = high need)	SCNS for partners and caregivers	[30]
SCNAT-IP	4: physical and psychological, hospital care, information and communication, and practical and cultural needs	26	Five-point scale (1 = no need/not applicable, 5 = high need)	Assessment tool for Indigenous Australians (based on modified SCNS-SF34)	[28]
CPNQ	5: psychosocial, health information, physical and daily living, patient care and support and interpersonal communication	76	Five-point scale (1 = no need/not applicable, 5 = high need)	Precursor questionnaire to the current SCNS; one of the first instruments to investigate SCNs	[18]
BR- CPNQ	5: psychological, health information, physical and daily living, patient care and support and interpersonal communication	52	Five-point scale (1 = no need/not applicable, 5 = high need)	Version of the CPNQ for breast cancer patients	[20]
CPNQ-SF	5: psychological, health information, physical and daily living, patient care and support and interpersonal communication	32	Five-point scale (1 = no need/not applicable, 5 = high need)	CPNQ short form in ambulatory cancer setting	[19]
CPNQ-revised	6 need categories: informational, practical, emotional, psychosocial, physical and spiritual	45	Five-point scale (1 = no need/not applicable, 5 = high need)	CPNQ revised version of SCNs for parents of children with cancer	[21]
CANDI	7: depression, anxiety, emotion, social, healthcare, practical and physical	39	Five-point scale (1 = not a problem, 5 = very severe problem); additional choices: ‘prefer not to answer’ or ‘do not know’	Cancer Need Distress Inventory based on biopsychosocial model	[31]
CARES	5 summary scales: physical, medical, marital, psychosocial and sexual	139	Likert 0–10 scale (0 = not at all and 10 = a great deal); patients can answer from a minimum of 93 items to a maximum of 132	First clinically relevant tool to assess rehabilitation needs and daily living problems of cancer patients	[17]
CaSUN	5: existential survivorship, comprehensive care, information, QOL and relationships	28	Three-point scale (met need, unmet need or total need)	Tool to assess cancer survivors’ unmet supportive care needs	[32]
CaSUN-Dutch	6: existential survivorship, comprehensive care, information, QOL, relationships and lifestyle and return to work	37	Three-point scale (met need, unmet need or total need)	Extended with five items on return to work and four on lifestyle, because these are prominent issues among cancer survivors, and they may also experience unmet needs in these domains	[34]
CaSUN-Chinese	4: information, physical/psychological, medical care and communication needs	20	Three-point scale (met need, unmet need or total need)	Added 11 items for women with breast cancer, then 18 items were eliminated, because <10% of women reported these items as unmet needs, and 7 items were omitted because of their low item total correlation	[33]
CaSPUN	7: information and medical care, socioeconomic issues, physical functioning, relationship issues, emotional issues, expectations of self and others, and life perspective	47 unmet needs items and 6 positive outcome items	For unmet need items: three-point scale (met need, unmet need or total need); for positive outcome items: four response options (‘yes, but I have always been like this’, ‘yes, this has been a positive outcome’, ‘no, and I would like help to achieve this’ or ‘no, and this is not important to me’)	Tool to assess unmet supportive care needs in partners of cancer survivors	[35]
SUNS	5 subscales: information, financial concerns, access and continuity of care, relationships and emotional health	89	Five-point scale (0 = no unmet need, 4 = very high unmet need)	Tool to evaluate unmet needs of adult cancer survivors who are 1 to 5 years post-cancer diagnosis	[36]
SUNS-SF	4 subscales: information, financial concerns, access and continuity of care and relationships and emotional health	30	Five-point scale (0 = no unmet need, 4 = very high unmet need)	Shortened form of SUNS: the emotional health and relationships domains loaded onto one factor	[37]
SUNS-hematological	5 subscales: information, financial concerns, access and continuity of care, relationships and emotional health	89	Five-point scale (0 = no unmet need, 4 = very high unmet need)	SUNS for hematological cancer survivors	[38]
NA-ACP	7: medical communication/information, psychological/emotional, daily living, financial, symptom, spiritual and social	132	Five-point scale (1 = no need/not applicable, 5 = high need)	Tool specifically designed to assess the needs of patients with advanced incurable cancer	[39]
NA-ALCP	7: medical communication, psychological/emotional, daily living, financial, symptom, spiritual/existential and social	38	Likert 0–10 scale (0 = not at all and 10 = a great deal)	Shortened version on NA-ACP specific for advanced lung cancer patients	[42]
SPEED	5: physical, spiritual, social, therapeutic and psychological	120	Likert 0–10 scale (0 = not at all and 10 = a great deal)	A palliative medicine needs assessment tool for patients with cancer in the emergency department	[44]
3LNQ (Danish)	3: problem intensity, problem burden and felt need	16	Problem burden (not at all to very much); Felt need (does not have a problem: no need; has a problem but does not want help: no need; met need; unmet need; partially unmet need)	Tool that measures 12 important needs with three different approaches (problem intensity, problem burden and felt need) when used as a supplement of EORTC QLQ-C30	[45]
CNAT (Korean)	7: healthcare staff, physical symptoms, psychological problems, information, social/religious/spiritual support, practical support and hospital facilities and services	51	Likert 0–10 scale (0 = not at all and 10 = a great deal)	CNAT aims to cover cancer patients’ needs in a comprehensive way throughout all phases of the cancer experience, from the diagnosis to recovery or palliative care, and to assess the general needs of cancer patients which are commonly applicable to a relatively vast majority of cancer types	[50]
CNAT-caregivers (Korean)	8: physical health, psychological needs, family/social relationships, healthcare staff-related needs, information/education needs, religious/spiritual support, hospital facilities and services and practical needs	41	Four-point scale (1 = no need, 4 = high need)	CNAT for cancer caregivers in patient–caregiver dyads	[51]
PNPC (Dutch)	Both the problem aspect and need for care aspect, ADL and IADL, physical symptoms, role activities, financial/administrative, social, psychological, spiritual, autonomy, problems in consultations, overriding problems in the quality of care, need for care aspect, concerning the general practitioners, concerning the specialist and informational needs	138	The PNPC asks 2 questions at each item: 1. Is this (item) a problem? yes/somewhat/no 2. Do you want (professional) attention for this (item)? yes/as much as now/no	Tool to assess needs in cancer patients in palliative care; it was developed to support the provision of care tailored to the specific demands of patients	[40]
ISQ (Greek)	2 subscales: disease and treatment and psychological	17	Three-point scale (I absolutely need to know; I would like to know; I do not want to know)	A questionnaire for the assessment of general and specific cancer patients’ needs for information regarding their disease	[46]
SST-IUPCN	5: extent of disease, performance status, prognosis, comorbidities and PC-specific problems	11	Total score ranges from 0 to 14	Screening tool for identifying unmet palliative care needs in cancer patients	[41]
NEQ	4: informative needs about diagnosis and prognosis, informative needs about exams and treatments, communicative needs and relational needs	23	Dichotomous (present vs. absent)	First Italian tool to evaluate needs of hospitalized cancer patients	[47]
NEQ	5: informative needs about diagnosis, prognosis and treatments, needs related to assistance/care, relational needs, needs for a psycho-emotional support and material needs	23	Five-point scale (0 = no unmet need, 4 = very high unmet need)	Differs from the first version in the response mode (5-point Likert vs. dichotomous)	[48]
SCC (French and English)	5 sections: social, nutritional, physical, pain and psychological	5 questions to assess social needs, SEFI to assess nutritional needs, 4 questions to assess physical needs, NRS to assess pain and GAD2 and PHQ-2 to assess psychological needs	An algorithm has been developed to generate alerts in the different sections of the SCC: social (patient living alone and needing assistance at home pr patient who has no social security coverage and/or no mutual insurance and/or has not lived in France for more than 3 months); nutritional (SEFI < 7); physical (fatigue restricting daily life activity or stopping physical activity after onset of the disease); pain (NRS ≥ 4); psychological (PHQ-2 or GAD2 ≥ 3)	The Supportive Care sCore (SCC) is a new screening tool developed to trigger alerts on major supportive care needs	[49]

**Table 2 ijerph-21-00215-t002:** Intervention options for supportive care needs.

Intervention Type	Reference Country	Intervention Purpose	Population Targeted	Description of the Intervention	Main Results
Implementation of smartphone and tablet interactive app	[64] Sweden	To investigate patients’ experiences with respect to an interactive app developed to help in managing symptom burden and sustaining self-care after pancreaticoduodenectomy	26 patients undergoing pancreaticoduodenectomy due to pancreatic or periampullary region cancer	Patients were provided with an interactive app (Interaktor) to monitor regular self-evaluations of symptoms, alert reports, access to self-care advice/websites for more information and perception of symptoms history; patients were requested to report symptoms daily for at least 4 weeks starting from the first day after discharge and up to 6 months after surgery or one week after ceasing adjuvant chemotherapy	Interaktor app was well accepted by patients as they received reassurance and support in self-care; patients reported lower symptom burdens and higher self-care activities The app enabled patients’ participation in care planning, facilitating person-centered care
Patient education and psychological support	[65] Taiwan	To investigate the effects of education and psychological support on anxiety, symptom burden, social support and unmet supportive care needs	80 female patients newly diagnosed with breast cancer, over 3 months after surgery undergoing chemotherapy	Patients received three individual face-to-face educational and psychological support sessions and two telephone follow-up sessions; educational sessions provided knowledge about disease and treatment (symptom burdens and side effects) and self-care (how to manage diet, hot flushes, loss of energy, etc.); psychological sessions offered stress-free time to express thoughts and feelings and enhance coping skills	Symptom distress and psychological, physical and practical supportive care needs were reduced; issues and needs about intimate and sexual relationships were not met
Telephone-based psychotherapeutic intervention	[66] Australia	To investigate the effectiveness of a telephone-based cognitive behavioral intervention on psychological outcomes and supportive care needs	163 distressed cancer patients (distress thermometer ≥ 4)	Patients received five sessions of a telephone-based cognitive behavioral therapy intervention (CancerCope) focused on the cancer journey, understanding stress, managing worry, tackling problems, improving well-being and moving forward, based on educational information, expert videos and stories/videos of 4 fictional characters	The intervention had no effects on symptom burden, supportive care needs, quality of life and post-traumatic growth, but only on psychological distress, cancer-specific distress and unmet psychological care needs
Combined intervention: nurse consultation and peer support	[67] Australia	To investigate the impacts of a combined nurse- and peer-led psychoeducational intervention in enhancing psychological distress, preparation for treatment, quality of life, psychosexual functioning, unmet supportive care needs and vaginal stenosis	319 gynecological cancer patients undergoing radiotherapy with curative intent	Four nurse-led face-to-face/telephone consultations plus four peer-led psychosocial telephone support sessions; nurse-led consultations focus on (1) radiation facility tour and consultation, (2) education on radiotherapy side-effects, use of vaginal dilator and on pelvic floor muscle exercises, (3) free discussion on issues and concerns about side-effects and psychosexual recovery and (4) free discussion on issues and concerns about self-care; peer-led psychosocial support aims to promote adherence and self-care strategies	Intervention effects on treatment preparation, sexuality/health system and information supportive care needs (no effect on psychological distress)
Complementary and alternative medicine (massage)	[68] USA	To investigate the effects of Swedish massage therapy on ailments and needs due to cancer-related fatigue	66 stage 0-III breast cancer patients, 3 months–4 years post treatment with or without ongoing chemoprevention	Massage therapy lasting 45 min; firstly, the subject was in a prone position and the therapist worked slowly down the body from the shoulders to the feet; then, the subject turned to the supine position and the therapist worked from the feet to the shoulders and head; effleurage, petrissage and tapotement techniques were included	The intervention produced significant relief in fatigue and quality of life improvements
Home-based, light-intensity physical exercise	[69] USA	To investigate feasibility and safety of home-based light-intensity exercise and its effect on the perceived self-efficacy regarding cancer-related fatigue self-management	7 patients received thoracotomy for early-stage, non-small cell lung cancer, transitioning from hospital to home	Six-week intervention of warm-up exercises, light-intensity walking and balance exercises in a virtual reality environment (Nintendo Wii Fit Plus); exercises increased in duration and intensity from week to week under nursing telephone control	The intervention was feasible, safe, well tolerated and accepted, and significantly increased perceived self-efficacy for fatigue self-management, walking, balance and functional performance
Implementation of electronic health records, accessible to patients	[70] USA	To investigate the effect of the integration of an electronic health record accessible to the patient to facilitate the report of patient-reported outcomes and supportive care needs	3521 oncology outpatients	Electronic assessment of patient-reported outcomes (including anxiety, depression, pain, fatigue, physical functioning and supportive care needs) using clinical thresholds for screener alerts	The implementation of electronic health records facilitated assessment and reporting of patient-reported outcomes and supportive care needs in oncology outpatient care

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
