# Peer review of "A Critical Overview of the Construct of Supportive Care Need in the Cancer Literature: Definitions, Measures, Interventions and Future Directions for Research"

_ijerph, 2024, doi:10.3390/ijerph21020215_

Round 1

Reviewer 1 Report

Comments and Suggestions for Authors

Dear Author,

Thank you for submitting your manuscript titled "A Critical Overview of the Construct of Supportive Care Need in the Cancer Literature: Definitions, Measures, Interventions and Future Directions for Research" for review. I have carefully reviewed the manuscript and would like to provide you with some constructive feedback.

Lack of emphasis on educational support in supportive care:
While your manuscript provides a comprehensive overview of the construct of supportive care need in the cancer literature, I believe there is a missed opportunity to discuss the importance of educational support within the realm of supportive care. Educational support plays a vital role in empowering patients and their families to actively participate in their care, make informed decisions, and improve self-management skills. I recommend expanding on this aspect by discussing the potential benefits and challenges of educational support interventions in the context of supportive care.

Insufficient specific guidance on supportive care interventions:
Although you have covered definitions, measures, and future directions for research in the construct of supportive care need, the manuscript lacks specific guidance or recommendations for supportive care interventions. Including practical and evidence-based recommendations for interventions would greatly enhance the applicability of your review. Consider providing examples of effective interventions, such as psychosocial support programs, symptom management strategies, or patient navigation services, and discuss their impact on patient outcomes. Additionally, you may want to highlight the importance of tailoring interventions to individual patient needs and preferences.

Overall, your manuscript provides a valuable critical overview of the construct of supportive care need in the cancer literature. However, addressing the aforementioned points would further strengthen the manuscript and make it more informative for readers. I encourage you to consider these suggestions during the revision process.

Thank you again for the opportunity to review your work. I look forward to seeing the revised manuscript.

Best regards,

Reviewer

Author Response

I thank the Reviewer for his/her precious revision. Please, see the attachment below.

Reviewer 2 Report

Comments and Suggestions for Authors

I want to thank the authors for bringing up this important public health issue, which can significantly supplement the literature.

I have a few minor comments.

Introduction: A lot of concepts are presented without citing a credible reference, particularly describing clinical and mental problems associated with cancer diagnosis. Moreover, this section looks a bit jumpy. The authors can improve the flow of the introduction section. 

Measures: A more detailed critique of the scale is needed.

conclusion and issues for the future: The conclusion is adequate. 

Author Response

(The authors gave the same response as above.)

Reviewer 3 Report

Comments and Suggestions for Authors

Title: A Critical Overview of the Construct of Supportive Care Need in the Cancer Literature: Definitions, Measures, Interventions and Future Directions for Research.

Summary: In this review article, authors have addressed the importance of supportive care for cancer patients, and how critical it is to have a well-defined mechanism/protocol for supportive care for their outcome. Authors have consolidated the current understanding of need for supportive care, and they have discussed the tools for care and intervention available to date.

Overall, I think the review focuses on a very important issue. It is well written and addresses all the important factor associated with cancer patients care and support.

I have no comments.

Paper can accepted in the current form.

Author Response

I thank the Reviewer his/her appreciation of the manuscript and the time spent in reviewing it.
